# An Overview of Clinical Examinations in the Evaluation and Assessment of Arterial and Venous Insufficiency Wounds

**DOI:** 10.3390/diagnostics13152494

**Published:** 2023-07-27

**Authors:** Szu-Han Wang, Victor Bong-Hang Shyu, Wen-Kuan Chiu, Ren-Wen Huang, Bo-Ru Lai, Chia-Hsuan Tsai

**Affiliations:** 1Department of Plastic and Reconstructive Surgery, Chang Gung Memorial Hospital, Keelung Branch, Keelung 204, Taiwan; shadowghost8@hotmail.com (S.-H.W.); vbshyu@gmail.com (V.B.-H.S.); fighter624@cgmh.org.tw (B.-R.L.); 2Graduate Institute of Biomedical Informatics, Taipei Medical University, Taipei 110, Taiwan; 3College of Medicine, Chang Gung University, Taoyuan 333, Taiwan; aaronhuang0327@gmail.com; 4Division of Plastic Surgery, Department of Surgery, Wan Fang Hospital, Taipei Medical University, Taipei 116, Taiwan; plastychiu@tmu.edu.tw; 5Department of Surgery, School of Medicine, College of Medicine, Taipei Medical University, Taipei 110, Taiwan; 6Division of Trauma Plastic Surgery, Department of Plastic and Reconstructive Surgery, Chang Gung Memorial Hospital, Taoyuan 333, Taiwan

**Keywords:** chronic wound, arterial insufficiency, venous insufficiency, clinical examination

## Abstract

Arterial and venous insufficiency are two major causes of chronic wounds with different etiology, pathophysiology, and clinical manifestations. With recent advancements in clinical examination, clinicians are able to obtain an accurate diagnosis of the underlying disease, which plays an important role in the treatment planning and management of patients. Arterial ulcers are mainly caused by peripheral artery diseases (PADs), which are traditionally examined by physical examination and non-invasive arterial Doppler studies. However, advanced imaging modalities, such as computed tomography angiography (CTA) and indocyanine green (ICG) angiography, have become important studies as part of a comprehensive diagnostic process. On the other hand, chronic wounds caused by venous insufficiency are mainly evaluated by duplex ultrasonography and venography. Several scoring systems, including Clinical–Etiology–Anatomy–Pathophysiology (CEAP) classification, the Venous Clinical Severity Score (VCSS), the Venous Disability Score, and the Venous Segmental Disease Score (VSDS) are useful in defining disease progression. In this review, we provide a comprehensive overlook of the most widely used and available clinical examinations for arterial and venous insufficiency wounds.

## 1. Introduction

Chronic wounds are defined as wounds that fail to heal in an orderly and timely manner. An acute wound will undergo four phases to achieve good healing: hemostasis, inflammation, proliferation, and remodeling [1]. If any phase is interrupted, the acute wound may become a chronic wound due to inappropriate healing. Globally, 1.67 out of every 1000 people suffer from chronic wounds [2]. Chronic wounds are a great burden to patients and society and are also challenging for wound care professionals.

The cause of wound healing disruption can be local or systemic. Local factors that influence healing include tissue hypoxia, infection, the presence of a foreign body, and venous insufficiency. Systemic factors include old age, stress, ischemic status, systemic diseases (e.g., diabetes mellitus and obesity), medication (e.g., glucocorticoid steroids), alcoholism, smoking, immunocompromised status, and poor nutrition [3]. Identifying these factors and making the correct diagnosis are the first steps in treating chronic wounds. In this article, we will review several different clinical examinations and diagnostic tools used in the diagnosis of several different types of chronic wounds, namely diabetic foot ulcers, arterial insufficiency, and venous insufficiency.

## 2. Arterial Ulcers and Diabetic Foot Ulcers

### 2.1. Overview

Arterial ulcers are mainly caused by reduced arterial blood supply, of which the most common etiology is peripheral artery disease (PAD, also called peripheral artery occlusive disease or PAOD) [4]. Peripheral artery obstruction is related to atherosclerotic disease. The symptoms of PAD can be silent or symptomatic and present in various ways. Lower extremity pain, claudication, non-healing wounds, and ulcer formation are the most common symptoms [5]. A total of 4.3% (95% CI 3.1% to 5.5%) of the population older than 40 are diagnosed with PAD. Among those aged 70 years and above, the prevalence rises to 14.5% (95% CI 10.8% to 18.2%) [6]. Although 75% of PAD patients are asymptomatic, there can still be severe manifestations of peripheral artery disease. PAD is associated with a 20% 1-year mortality rate and a 1-year limb loss rate of 20% [7,8,9]. Hence, leg ulcer formation is considered to have a poor prognosis with a 50% amputation rate and a 48% 1-year mortality rate [10].

A foot ulcer is also one of the major causes of morbidity in diabetes mellitus (DM) patients. In patients with either type 1 or 2 DM, the lifetime prevalence of foot ulcers can be up to 34% [11]. The main causes of foot ulcers are peripheral neuropathy and arterial insufficiency, which can be present in over 80 percent of diabetic patients with foot ulcers. Insensibility to pain and decreased perception of pressure occurs, while impaired microcirculation causes a breakdown of the skin and leads to ulcer formation. The process of wound healing is difficult due to a lack of adequate blood flow and concomitant deep tissue or bone infection. Although percutaneous angioplasty with optional stenting has become an efficient intervention to deal with the situation of arterial insufficiency in the lower limb, complications such as hematoma, thrombosis, pain, and subintimal dissection should still be considered during management [12,13,14]. Moreover, the risk of restenosis should also be considered, especially since DM itself is a risk factor for restenosis [15,16,17,18]. Anatomical deformity is also seen in diabetic patients, including hammertoes, loss of arch, and rocker bottom feet associated with Charcot’s foot. Abnormal plantar pressure redistribution can make the situation even worse [19].

Inadequate treatment of diabetes foot ulcers can lead to severe consequences, such as major limb amputation or even death. The 5-year mortality rate for patients who have limb amputation after diabetic foot ulcer infection is 50% [20]. With proper care of diabetic foot ulcers, up to 70% of amputations can be prevented [21]. As a consequence, early identification and providing adequate treatment to patients is important and necessary. We will review several methods that assist in the detection and early diagnosis of arterial ulceration and diabetic foot ulcers.

### 2.2. Pulsation and Hand-Held Doppler

Direct palpation of pulsation is part of a thorough physical examination and can be easily performed in an office setting. All major arteries, including femoral, popliteal, posterior tibial, and dorsalis pedis pulses should be palpated during the physical examination and simultaneously compared with the ipsilateral radial artery and contralateral corresponding site [7,22]. The intensity, rate, rhythm, and presence of blood vessel tenderness, tortuosity, or nodularity should be checked. The pulse can be classified as absent, diminished, or normal (a graded scale of 0–2) or even detailed as no palpable pulse, faint but detectable pulse, slightly more diminished pulse than normal, normal pulse, and bounding pulse (a graded scale of 0–4+) [7,23].

Hand-held Doppler is the most frequently used noninvasive vascular examination in clinical practice. It is a portable ultrasound device designed to detect pulsation that cannot be palpated manually. Several arteries should be examined during the assessment, as mentioned below.

The dorsalis pedis artery, a continuum of the anterior tibial artery, can usually be palpated lateral to the extensor hallucis longus tendon (or medial to the extensor digitorum longus tendon) on the dorsal surface of the foot. However, dorsalis pedis can only be palpated in 78% of healthy extremities. Five percent of dorsalis pedis pulsation can still be missing when using hand-held Doppler [24]. Although there is low specificity, the abnormality of dorsalis pedis pulsation is still the most prevalent finding in peripheral arterial disease [25].

On the contrary, the absence of a posterior tibial artery (PTA) pulse always indicates an abnormality of the artery [25]. The PTA branches from the popliteal artery and supplies the posterior compartment of the lower leg. It passes behind the medial malleolus and underneath the flexor retinaculum, where it can be palpated. After passing the medial malleolus, the PTA gives off two branches, the medial and lateral plantar artery. Abnormality of the PTA means both medial and lateral plantar arteries may be jeopardized, and the plantar circulation can only be supplied from reverse flow through various arterial–arterial anastomosis, such as cruciate anastomosis [26].

The popliteal artery lies in the popliteal fossa, formed by the semimembranosus muscle and biceps femoris muscle proximally, and gastrocnemius muscle medial and lateral head distally. The popliteal artery originates from the superficial femoral artery and branches into the anterior and posterior tibial arteries distally. Although there is some collateral arterial circulation, the popliteal remains the only major artery that runs through the knee joint [27]. Palpating popliteal artery pulsation can help in recognizing varying levels of arterial occlusion if other distal artery pulsations are not palpable.

The femoral artery, as a continuation of the external iliac artery, after it passes the inguinal ligament, is the main blood supply to the lower limb. It bifurcates into the superficial and deep femoral arteries. The femoral artery is usually palpated at the femoral triangle. The femoral triangle is bounded superiorly by the inguinal ligament, laterally by the medial border of the sartorius muscle, and medially by the medial border of the adductor longus muscle. Halfway between the pubic symphysis and anterior superior iliac spine and one inch caudal to the inguinal ligament is usually the point where the femoral artery pulse can be palpated. Femoral artery pulse can be palpated when the patient’s systemic systolic blood pressure is over 50 mmHg [28].

### 2.3. Ankle Brachial Index (ABI)

The ankle brachial index (ABI) or ankle brachial pressure index (ABPI) is one other measurement that can be performed in the outpatient clinic office. It is non-invasive and inexpensive, which makes it the most widely used measurement in daily practice. The American Heart Association also recommends using the ABI for screening high-risk patients in the clinical setting for PAD [29].

The ABI measures the systolic blood pressure of both ankles and the brachial systolic blood pressure of both arms and then divides each ankle’s systolic blood pressure by the higher systolic brachial blood pressure. Classically, blood pressure is measured by checking the pulsation distal to an inflatable cuff under different degrees of inflation. As the cuff deflates, the pressure measured as the first pulsation is detected is determined to be the systolic pressure. Stethoscopes and Doppler probes can be used to detect the pulse, although recently designed oscillometric and photophlethysmographic devices can help to more accurately detect the pulsation of vessels [30].

A normal ABI range is between 1.00 and 1.40, while an ABI ≤ 0.90 is defined asabnormal and values >1.40, which indicate noncompressible arteries. ABI values of 0.91 to 0.99 are considered borderline [8,31]. Xu et al. have published a review article reporting high specificity (83.3–99.0%) and accuracy (72.1–89.2%) when using ABI ≤ 0.90 as the criterion for detecting a greater than 50% stenosis of the lower limb artery [32]. However, the sensitivity of the ABI in detecting stenosis is quite variable, ranging from 15–20% [33] to 70.6% [34]. According to a study conducted by Vega et al. in 2011, using an oscillometric ABI can achieve a sensitivity of 97% (95% CI 93% to 99%) and a specificity of 89% (95% CI 67% to 95%). A manual Doppler ABI can achieve a sensitivity of 95% (95% CI 89% to 97%) and a specificity of 56% (95% CI 33% to 70%) [35].

It should be noted that both a low (<1.00) and high (≥1.40) ABI may indicate a significant abnormality. A low ABI indicates arterial stenosis of the lower extremities, which is also associated with cardiovascular risk factors such as hypertension, dyslipidemia, and diabetes mellitus [36,37]. Additionally, a low ABI also demonstrates a strong relationship with the prevalence of coronary artery disease and cerebrovascular disease [38]. Criqui et al. have reported that both lower and higher ABIs were associated with incident cardiovascular events [39]. In the situation of an abnormally high ABI, calcification of the vascular wall was the most likely cause, which could be associated with medial carcinosis, diabetes mellitus, and end-stage renal diseases [40,41]. We may miss the diagnosis of PAOD by using the ABI only if occlusive lesions of the lower extremities happen simultaneously with arterial wall calcification. Maruhashi et al. have reported two cases of peripheral artery disease; however, the ABI was found to be normal in both cases [42]. As a result, ABI examination may be unreliable in patients with a high possibility of vessel wall calcification, and alternative examinations should be considered to confirm the diagnosis.

Above all, despite the efficiency of using the ABI in the diagnosis and evaluation of artery condition and quality, it shows no correlation to wound healing prediction and prognosis of chronic wounds [43].

### 2.4. Transcutaneous Oxygen Measurement (TcPO2)

Transcutaneous oxygen measurement (TcPO2) is another non-invasive method that measures the amount of oxygen in capillaries. TcPO2, also known as transcutaneous oximetry, places an electrode on the skin surface and heats the skin gently to increase blood flow into the area. By measuring the oxygen flow in the capillaries, it can provide an indirect measurement of blood flow to the tissue [44].

TcPO2 is used as one of the factors to predict wound healing after proper treatment. The result of TcPO2 is either shown as O2 pressure or ratio to normal tissue. The normal value of foot TcPO2 during the supine position is approximately 60 mmHg, and the normal chest/foot ratio is approximately 0.9 [45].

Caselli et al. measured diabetic foot ulcer patients’ TcPO2 after receiving percutaneous transluminal angioplasty [46]. Of the recruited 43 patients receiving revascularization, 20 of them had failed revascularization. TcPO2 progressively improved after successful revascularization and reached peak value at 4 weeks after the procedure. On the other hand, unsuccessful revascularization only showed a slight improvement in TcPO2. The percentage of patients with TcPO2 above 30 mmHg increased from 38.5% to 75% at 3–4 weeks after the procedure [46].

Ladurner et al. recruited 141 diabetic patients with non-palpable pedal pulses. They subgrouped patients according to their TcPO2 into three groups: TcPO2 < 20 mmHg, TcPO2 20–40 mmHg, and TcPO2 > 40 mmHg. After one year of proper wound treatment, patients in the group with TcPO2 > 40 mmHg had a significant possibility of wound healing compared to patients with TcPO2 < 20 (73% vs. 48%, *p* = 0.008). However, there was no significant difference in wound healing when comparing TcPO2 20–40 mmHg patients with either TcPO2 > 40 patients or TcPO2 < 20 patients [47].

The power of TcPO2 in predicting diabetic foot ulcer wound healing is also demonstrated by Zubair’s study in 2019, which enrolled 192 patients with diabetic foot ulcers [48]. One hundred and nine patients achieved ulcer healing with intact skin, while the other eighty-three patients had either mild improved ulcer healing or impaired ulcer healing. They compared patients’ TcPO2 and found that patients with healed ulcers had slightly higher TcPO2 (50.834 ± 18.77 vs. 50.60 ± 18.76, *p* = 0.049). They concluded that TcPO2 has a slightly positive correlation with wound healing ability [48].

Although the correlation between TcPO2 and wound healing is still controversial based on the literature review, it remains a useful tool to provide helpful information for the diagnosis of PAD and the management of chronic wounds.

### 2.5. Absolute Toe Systolic Pressure

Although the ABI has good specificity in detecting peripheral artery stenosis, the presence of arterial calcification known as Mönckeberg’s sclerosis leads to ABI variation in sensitivity when screening for stenosis [49]. Toe systolic pressure has been available since early the 1930s and is less affected by arterial calcification [50].

The great toe is usually the choice of measurement in toe systolic pressure because of its size. However, the great toe frequently has ulcers or has been amputated. In such cases, second-toe systolic pressure measurement is a reliable alternative. Bhamidipaty et al. demonstrated that second-toe systolic pressure is highly correlated to first-toe systolic pressure (correlation coefficient (r) = 0.908 and 0.877 to 0.931 under a 95% confidence interval) [51].

The Intersociety Consensus for the Management of Peripheral Arterial Disease defined the normal range of the toe systolic pressure in healthy people as approximately 30 mmHg lower than the systolic pressure obtained from the ankle. When toe systolic pressure is under 30 mmHg or the toe brachial index (TBI) is less than 0.7, it is considered abnormal [9].

Clinical usage of toe systolic pressure as the sole evidence of peripheral artery occlusion is questionable. A clinically significant margin of error was found in toe systolic pressure measurement. Considering that the toe systolic pressure may range from 40 to 90 mmHg and around ±26 mmHg of fluctuation error may be found during measurement, toe systolic pressure may not be a reliable vascular assessment option [49]. The same condition is also observed in the toe brachial index.

There is some evidence that the usage of toe systolic pressure as a predictive factor of diabetic foot ulcer healing is feasible. Tay et al. conducted a meta-analysis of the sensitivity and specificity of toe pressure in the prediction of diabetic foot ulcer wound healing [52]. A total of 909 patients in 8 studies were included in the analysis. A sensitivity of 0.86 (95% CI: 0.82–0.89) and specificity of 0.58 (95% CI: 0.52–0.63) were found when using toe blood pressure 30 mmHg as the cutoff point in predicting wound healing. But the studies included were relatively old; five of them were from the 1980s and two were from the 1990s. With the availability of automatic photoplethsmography in measuring toe blood pressure [53], more recent clinical trials are needed to determine the toe blood pressure’s diagnostic value or wound healing prediction power.

### 2.6. Arterial Doppler Studies

During the past five decades, ultrasound technology has advanced dramatically and has been widely adopted in detecting vascular disorders [54]. In addition to detecting the presence of a pulse, an arterial Doppler study can also provide either the audio or visual waveform of the pulse. A multiphasic waveform, including biphasic or triphasic sounds, tends to be a normal waveform. In contrast, a monophonic waveform is usually considered abnormal, which may indicate a decreased elasticity of the vessel.

The ability to measure the volumetric blood flow in a non-invasive way without contrast agents is the major advantage of arterial Doppler exams [55]. The measurement can not only be performed over larger arteries, such as the common femoral artery, but also on smaller arteries, like the dorsalis pedis artery. Holland et al. performed an arterial Doppler exam on 42 patients and documented the blood flow volume in different arteries [54]. They recorded an average flow velocity of 284 ± 119 mL/min in the common femoral artery, 152 ± 66 mL/min in the superficial femoral artery, 72 ± 34 mL/min in the popliteal artery, and 3 ± 4 mL/min in the dorsalis pedis artery. However, the blood flow reported in their study is less than in other studies. Lewis et al. and Hussain et al. also performed measurements on the common femoral artery and found flow velocities of 350 ± 141 and 359 ± 114, respectively [56,57]. In addition to being used as a diagnostic tool, the measurement can also be used for post-angioplasty outcome assessment [58].

On the contrary to musculoskeletal ultrasonography, which is an operator-dependent technique, arterial Doppler studies do not exhibit this issue. Tehan et al. conducted a survey comparing variability among different operators when performing an arterial Doppler exam [59]. They recruited 63 podiatrists to conduct an arterial Doppler exam and found excellent outcomes in both the inter- and intra-rater reliability of visual Doppler waveform interpretation.

Arterial Doppler studies can also help predict the risk of amputation in diabetic foot ulcers. In Barberan et al.’s study, arterial obstruction found in the Doppler study has an odds ratio of 12.50 (95% CI: 1.42–66.67; *p* = 0.003), resulting in amputation [60]. They reviewed 78 patients with acute diabetic foot infections and 26 of them resulted in amputation. Among the amputees, 84.6% were diagnosed with an obstruction on the arterial Doppler, and 15.4% were diagnosed with stenosis. In the limb salvage group, only 21.2% were classified as obstruction and 57.7% as stenosis. Tsai et al. also found that monophonic waveforms found in the dorsalis pedis artery or posterior tibial artery via a Doppler scan indicated a higher possibility of amputation in dialysis patients (odds ratio: 7.61) [61].

As a non-invasive examination, an arterial Doppler study can also serve as a screening test before definitive angiography or even angioplasty. Elgzyri et al. reviewed 166 patients receiving a Doppler artery study [62]. Fifty-five patients were recommended to receive angioplasty, with forty-two (76%) of them eventually receiving angioplasty and ten (18%) of them receiving diagnostic angiography only. Of the 64 patients recommended for diagnostic angiography, 23 (36%) received angioplasty, 10 (16%) had angiography followed by reconstructive surgery, and 24 (37.5%) had angiography only. All of the patients recommended for conservative treatment followed the recommendations.

### 2.7. Arteriography

While the various methods discussed above possess diagnostic power and wound healing prediction value in PAD and diabetic foot ulcers, arteriography remains the gold standard in diagnosis. There are several ways to obtain arteriography, including digital subtraction angiography (DSA), magnetic resonance angiography (MRA), and computerized tomographic angiography (CTA).

Digital subtraction angiography (DSA) is the gold standard of contrast arteriography for the assessment of lower extremity PAD. Most diagnostic studies are designed based on comparisons with DSA [63]. DSA is irreplaceable for its ability to provide real-time dynamic arteriography, even in distal small-caliber arterioles. Additionally, in a hybrid operation room setting, intravascular angioplasty or embolectomy can be performed simultaneously [64].

There are still several disadvantages of DSA. Complications resulting from the arterial access site and contrast agent reaction are the major concerns when performing DSA. Clinically significant bleeding, painful hematomas, and pseudoaneurysms are seen in femoral access sites, especially in patients under treatment with antithrombotic agents [65]. Up to 11.5% of patients are reported to have femoral access site complications when receiving endovascular procedures from the femoral artery [66]. Advanced age, a procedure duration greater than 45 min, below-the-knee procedures, a lower BMI, uncontrolled hypertension, and impaired plasmatic coagulation are found to be risk factors for access site complications [66]. The hyperosmolarity of the contrast agent may cause toxic side effects including nausea, vomiting, and pain in the area being studied. Allergic reactions can also be seen with minor or severe reactions, such as anaphylactic shock [67]. Nephrotoxicity from contrast agent use is another major concern in patients with diabetes mellitus or renal insufficiency. Pre-existing renal dysfunction is the major risk factor for developing contrast-induced nephropathy (CIN) in patients [68]. CIN occurs in 1% to 2% of patients with normal renal function. In patients with serum creatinine levels between 1.3 and 1.9 mg/dL, the incidence increases to 10%, further increasing to 62% in patients with creatinine levels greater than 2 mg/dL [68,69].

Magnetic resonance angiography (MRA) is one of the alternative angiography imaging modalities available. An MRA can be performed either with or without contrast enhancement. Contrast-enhanced MRA has a superior image resolution and is widely used in current practice [70]. Due to technological advances, contrast-enhanced MRA can be performed with either a single dose intravenous injection of a contrast medium [71] or a two-step injection technique for higher resolution quality [72]. Intravenous gadolinium-based agents are the main contrast agents used in MRA. Gadolinium chelates contrast medium for MRA in patients with normal renal function and is considered safe, with no CIN reported [73,74]. However, the safety of gadolinium-based contrast in renal-impaired patients is still under debate [75,76,77]. Nephrogenic systemic fibrosis, characterized by skin thickening and hyperpigmentation of the trunk and extremities, is reported in patients with chronic renal failure receiving gadolinium contrast agents. Nephrogenic systemic fibrosis can have severe effects, such as organ fibrosis or even death [64]. Furthermore, although MRA can provide a large volume of images over a long distance, image quality can easily be jeopardized by patient motions or breathing during the long acquisition time.

Computerized tomographic angiography (CTA) is a less time-consuming and less invasive option to obtain the arteriogram of the lower extremities (Figure 1). Easily performed while possessing high sensitivity and specificity, the CTA is one of the most practical exams in lower extremity vessel assessment. In one meta-analysis article conducted in 2009 [78], CTA provided a sensitivity of 95% (95% CI, 85–99%) and a specificity of 91% (95% CI, 79–97%) in detecting tibial arteries with greater than 50% stenosis or occlusion. It has an overall sensitivity of 95% (95% CI, 92–97%), and specificity was 96% (95% CI, 93–97%) in detecting greater than 50% stenosis or occlusion between the aorta–iliac region to the distal runoff of tibial arteries.

There are still some shortcomings of CTA. In patients with atherosclerotic calcification, the degree of stenosis may be overestimated [79]. In this scenario, the advancement of the calcium-subtracted technique in dual energy-CTA may help improve the accuracy and specificity [80]. It is also difficult to measure the entire picture of blood supply in a single phase of arteriography because the blood supply may be altered by collateral perfusion and microvascular events. Emerging techniques of perfusion examination may play an important role in addressing the limitation of arteriography [81]. Moreover, hazardous radiation exposure is another known drawback of CTA. The 70 kV tube voltage technique was developed to address this issue, with less contrast medium and radiation exposure dose. It has a mean effective radiation dose of 1.94 ± 0.21 mSv, while conventional digital subtraction angiography has an average effective dose of 4.41 ± 0.64 mSv [82]. With current developments, CTA has become an accurate and safe method to evaluate lower extremity arteries.

### 2.8. Other Emerging Methods

Indocyanine green (ICG) angiography was first introduced in the 1960s [83] and is widely used in ophthalmology. It has been used to detect peripheral vascular insufficiency in recent decades [84]. Indocyanine green is a water-soluble, non-radioactive, and non-toxic contrast agent and can be detected with infrared detecting devices. It can provide regional perfusion status and is widely used in microsurgery flap monitoring.

When ICG is used for peripheral vascular examination, 0.1 mg/kg of a 0.1% ICG solution bolus infusion is administrated intravenously, immediately followed by 10 mL normal saline. The room light is dimmed, and the limb photo is taken by an infrared camera. A high-intensity contrast-like appearance will be seen within seconds if the limb perfusion is good. This tool has proved to be effective in predicting diabetic foot ulcer wound healing [85]. More clinical trials are needed to determine the sensitivity and specificity of the test.

Laser speckle contrast imaging (LSCI) is another technique developed recently. It was first introduced in the 1980s. LSCI detects optical speckle patterns generated by the motion of red blood cells in the skin and provides information regarding microcirculation. Clinically, it has been considered an effective method to distinguish non-ischemic, ischemic, and critical-ischemic patients [86]. Similarly, more clinical trials are needed to determine the efficacy of the technique.

There are also several innovative techniques based on computed tomography imaging and magnetic resonance imaging that could improve the precision of tissue perfusion evaluation [81,87]. Computed tomography perfusion imaging is a technological advancement that allows tissue perfusion to be quantified by reconstructing the axial and temporal images [88,89,90,91]. Dynamic contrast-enhanced MRI examines tissue perfusion by acquiring baseline images without contrast and images during and after contrast administration [92,93]. The evaluation of T1 phase shortening after contrast administration is the main parameter for analysis and demonstrated great capability in discriminating PAD patients from healthy individuals [94]. Arterial spin labeling MRI is a non-contrast technique that measures tissue perfusion by labeling arterial water protons with radiofrequency pulses [95,96]. Blood oxygenation level-dependent MRI (BOLD-MRI) was first developed for brain activity mapping and is also known as the main technique used in functional MRI. BOLD-MRI measures tissue perfusion by the paramagnetic effect of deoxyhemoglobin. The difference in T2 signal intensity is the main analyzed outcome for tissue perfusion [81,87,97,98,99]. These new techniques mentioned above have the potential to improve the limitations of traditional examinations.

## 3. Venous Ulcer

### 3.1. Overview

Venous ulceration is the most common cause of lower leg ulcers, accounting for 70–80% of all lower leg ulcers [100]. It also serves as a clinical presentation of chronic venous insufficiency, which is present in up to 40–50% of the general population and is more commonly seen in females [101,102]. The etiology of chronic venous insufficiency can be classified as congenital, primary, or secondary. Although there are many associated pathologies related to venous ulcers such as endothelial dysfunction, deranged lymphatic function, associated arterial occlusive disease, joint disorders, and metabolic disturbances, venous reflux and obstruction are the main mechanisms leading to venous hypertension [103,104,105]. Venous reflux and obstruction can result from venous valve dysfunction, venous thrombosis, and phlebitis-related venous problems [103,104,105]. Although reflux alone is responsible for primary chronic venous insufficiency, secondary chronic venous insufficiency often results from a combination of both obstruction and reflux. Patients with combined chronic reflux and obstruction have the highest incidence of skin changes and ulceration [103,106].

There are several theories regarding pathophysiology following venous hypertension. Browse and Burnand’s theory focuses on the increased intraluminal pressure in the capillary bed [107]. Falanga and Eaglstein presented the trap hypothesis, theorizing that macromolecule leakage trapped growth factors and hemostatic substances, impairing the repair process [108]. Claudy et al. and Powell et al. revealed the influences of TNF-alpha in venous ulcer formation [109,110]. Poor venous drainage and resulting venous hypertension increase transmural pressure in postcapillary vessels, producing skin capillary damage, fluid exudation with associated leakage of inflammatory proteins, edema, and tissue malnutrition, which then favor inflammation, infection, thrombosis, tissue necrosis with lipodermatosclerosis, and eventual ulceration [103,106].

History taking and physical examination serve important roles in the diagnosis of venous ulcer. It is helpful to recognize the characteristics of venous ulcer wounds such as irregular wound edge, granulation, pigmentation, dermatitis, and peri-wound edematous [111,112]. However, physical findings can provide little clue to the presence, location, extent, or severity of venous valvular incompetence or venous obstruction [112]. Further characterization of the underlying anatomy and pathophysiology is crucial for successful therapy [103]. Risk factors such as age, being a female, a history of lower leg trauma, varicose vein, venous diseases, and thromboembolism, which could increase the risk of venous ulceration formation, should also be identified [102,105]. Clinical history should focus on the signs and symptoms that elicit a venous cause of the ulcer. These criteria may include leg heaviness and cramps, swelling, leg pain after ambulating that is relieved with rest and elevation, and a history of deep venous thrombosis and ulceration [103,106,112,113].

### 3.2. Scoring System of Clinical Venous Disorders and Venous Ulcers

There are several scoring systems that evaluate the severity of chronic venous insufficiency including Clinical–Etiology–Anatomy–Pathophysiology (CEAP) classification, Venous Clinical Severity Score (VCSS), Venous Disability Score, and Venous Segmental Disease Score (VSDS) [114,115]. First introduced in 1993, CEAP classification is the most widely adopted scoring system and has been updated in the following two decades [115,116].

CEAP classification includes four categories (clinical signs, etiology, anatomy, pathophysiology) in the assessment of clinical venous disorders. Clinical signs include visible clinical manifestations of venous disorders. For example, healed venous ulcers are categorized as C5, and active or recurrent venous ulcers are categorized as C6 or C6r. Etiology is divided into primary, secondary, congenital, and combined etiologies. The anatomy component describes the venous system where venous disorders occurred (superficial, deep, and perforator veins). The category of pathophysiology designates the presence or absence of venous reflux and/or obstruction [114,115,116,117,118].

The Venous Clinical Severity Score (VCSS) is based on physician-determined and patient-reported elements. Pain, varicose veins, venous edema, pigmentation, inflammation, induration, number of active ulcers, duration of active ulcers, size of active ulcers, and compliance with compression therapy are the parameters, with numerical grading from zero to three (none to severe) [114,119]. The Venous Disability Score (VDS) is a simple grading system based on the symptoms and physical limitations of the patient. The Venous Segmental Disease Score (VSDS) incorporates the anatomical and pathophysiological components of CEAP classification to evaluate the clinical condition of chronic venous disorders [114,116].

### 3.3. Duplex Ultrasonography

Duplex ultrasonography is a combination of B-mode echography and Doppler sonography. It provides a non-invasive, convenient, and effective diagnosis of venous disease, and has been frequently used in the diagnosis and evaluation of deep vein thrombosis, varicose veins, and chronic venous insufficiency [120,121]. Duplex ultrasonography provides information on both the direction and velocity of flow. It provides more information than vascular ultrasound alone, while simultaneously examining the etiological, anatomical, and pathophysiological aspects of chronic venous insufficiency and also providing real-time guidance if surgical intervention is to be performed.

The lower extremity venous system can be divided into superficial, deep, and perforator groups. In general, different frequency probes can be chosen to fit the appropriate depth of the vein. The 5 to 8 MHz linear probe is recommended for deep vein evaluation, while the 10 to 12 MHz linear probe is more suitable for superficial vein assessment. However, 3 to 5 MHz curvilinear probes should be considered in large-diameter legs or edematous limbs, which can provide better ultrasound wave penetration [122]. The assessment of duplex ultrasonography will focus on direction rather than velocity. Based on pathophysiology, venous reflux is an important factor in chronic venous diseases. Lurie et al. showed good precision and repeatability in the detection of venous reflux under specific standard protocols [123]. The standardized protocol is necessary because the patient’s posture and the timing of the procedure during the day show a significant influence on the reliability of the examination. Several studies have emphasized that the standing position with standard protocol helps achieve better accuracy of duplex ultrasonography in the diagnosis of venous reflux and chronic venous insufficiency [124]. Performing duplex ultrasonography tests in the morning in the standing posture may achieve the best precision and repeatability [125].

A full examination of duplex ultrasonography in the lower extremities extends to proximal systems, such as the iliac veins and obturator veins. The examination begins with the saphenofemoral junction (SFJ), and several relevant veins with their associated branches should be examined, such as the great saphenous vein (GSV), anterior accessory saphenous vein (AASV), and small saphenous vein (SSV) [126]. Comprehensive examination should start from the location, diameter, and function of each vein to evaluate the superficial varices, the competence of valves, and the evidence of previous venous thrombosis. The great saphenous vein (GSV) is the most examined vessel in the evaluation of the lower extremity. There are several variations related to the GSV, such as duplicated GSV, single GSV with a large subcutaneous tributary, or GSV with an anterior accessory saphenous vein (AASV); hence, tracing the venous course provides helpful assessment in varicose vein and elicits the status of reflux. AASV, which divides from the GSV right after the saphenofemoral junction, is often responsible for varicose veins [127].

Within existing literature, duplex ultrasonography demonstrates a sensitivity of 56% to 100% and a specificity of 77% to 100% in the diagnosis of upper extremity deep vein thrombosis [128]. Goodacre et al. also reviewed a hundred cohort studies comparing duplex ultrasonography with venography for the diagnosis of deep vein thrombosis. The study showed that duplex ultrasonography had a pooled sensitivity of 96.5% (95.1 to 97.6) for proximal DVT and a relatively poor sensitivity of 71.2% (64.6 to 77.2) for distal DVT, with a specificity of 94.0% (92.8 to 95.1) [129]. In general, duplex sonography is a convenient and useful tool for the evaluation of chronic venous insufficiency and associated etiologies (Figure 2 and Figure 3).

### 3.4. Venography (Venogram)

Venography is a contrast-based, invasive examination that offers a comprehensive visualization of the venous system. Ascending and descending venography are the primary methods of investigation and can be performed with additional pressure gradient measurements. In addition to immediate visualization of the lower extremity venous system, these methods combined allow for information on luminal diameter, venous flow and washout, thrombus characterization, valvular damage, pressure gradients, and other anatomical pathologies. Another potential benefit of venography is the opportunity for treatment at the time of examination. Disadvantages include limited ability to evaluate the extravascular tissue, limited range of differential diagnosis, a lack of consensus on hemodynamically significant pressure gradients, and the possibility of incomplete assessment due to contrast load, access, or technical difficulties [113].

Venography is performed under initial ultrasound guidance to determine vein access. A sheath is inserted proximal or distal to the suspected pathological region. An ascending venogram is performed by injecting contrast into a vein distal to the region of interest. For example, contrast injected into a dorsal vein of the foot can be directed into the deep venous system with the help of an ankle tourniquet (preferentially occluding the superficial system). This can demonstrate collateral venous pathways, regions of occlusion, valvular damage, perforating vein incompetence, and other critical findings. This can be extremely valuable as an adjunct to non-invasive testing results in patients with suspected lower extremity venous disease. The superficial system and perforating systems can also be evaluated with ascending techniques [103].

Descending venography is usually performed with the patient in a standing position using a tilting table. The main goal is to demonstrate venous reflux and points of leakage (from proximal to distal, or deep to superficial). Morphology of the venous system can also be visualized with this technique. The cannula can be introduced through the femoral or popliteal vein, proximal to the site of interest. Contrast injection then assists in evaluating the extent of the reflux, altered venous anatomy, and competence of junctions and valves. Results may help to determine a roadmap for venous system reconstruction through stenting procedures [103].

Due to the risks associated with invasive testing, contrast venography has largely been replaced by duplex ultrasonography for most clinical scenarios. Risks associated with venography include complications associated with the procedure itself, such as bleeding, thrombosis, tissue injury, and infection, and complications associated with contrast use, such as contrast media allergy and acute renal failure [130]. However, venography is still considered the gold standard in the detection of chronic venous obstruction, although it lacks quantitative results [106]. Precise anatomical visualization via venography is a valuable tool when duplex ultrasound leads to equivocal findings due to technical difficulty or insufficient resolution. Venography may also be required for surgical planning, especially in difficult or re-operative cases.

## 4. Conclusions

An overview of the common diagnostic procedures and their significance has been provided in this review. A precise diagnosis of the etiology and underlying factors leading to chronic wounds is critical to the therapeutic planning of these difficult pathologies. A combination of accurate patient history taking, detailed physical examination, and suitable diagnostic procedure selection will help the physician select the most effective treatment for the underlying condition. There are still limitations during wound assessment diagnostics, such as providing cost-effective instruments with bedside availability (ease of access). Furthermore, some invasive techniques may not provide enough information about tissue perfusion and vessel conditions, which weighs against invasive risk. New techniques that have become available recently still require validation by rigorous research under appropriate methodology. In our opinion, the future direction of diagnostic tools for chronic wound assessment will focus on examinations that are less invasive, cost-effective, and available at the bedside. The accuracy of the currently available technologies can also be improved by conducting further studies.

## Figures and Tables

**Figure 1 diagnostics-13-02494-f001:**
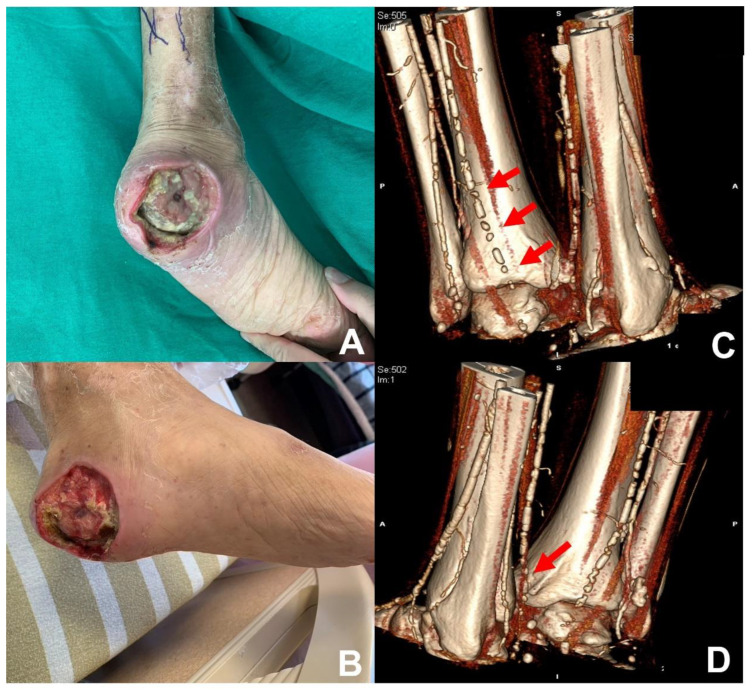
**Arterial ulcer.** A 46-year-old male with underlying diseases of type 2 diabetes mellitus, hypertension, peripheral arterial diseases, and end-stage kidney diseases under regular hemodialysis showed a chronic wound in the left heel. (**A**,**B**) Arterial ulcer at the left heel with necrotic tissues and bone exposure. (**C**,**D**) Multiple severe stenosis at the left posterior tibial artery were found under computed tomography angiography (**red arrow**).

**Figure 2 diagnostics-13-02494-f002:**
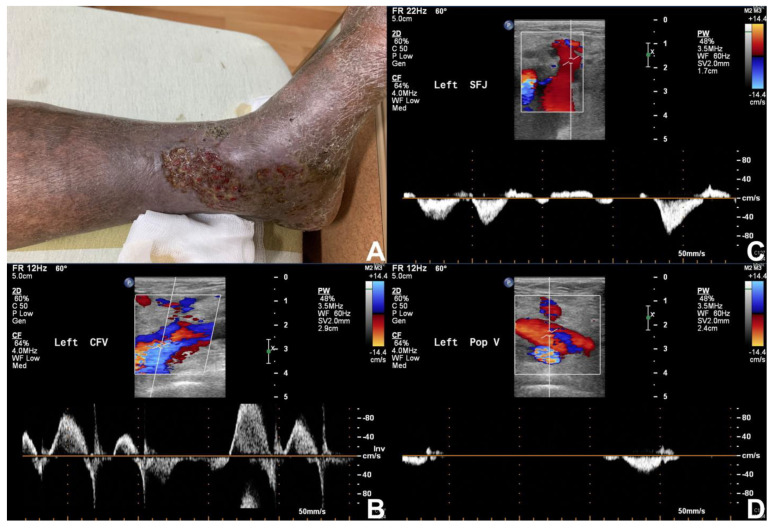
**Venous ulcer.** A 66 years old male with decompensated heart failure presented with chronic wound at medial side of left lower leg. (**A**) Venous ulcer at medial side of left lower leg with necrotic tissues. (**B**–**D**) Severe reflux and venous insufficiency of left sapheno-femoral junction (SFJ), common femoral vein (CFV) and left popliteal vein in the examination of duplex ultrasonography. Varicose vein was also noted during the examination.

**Figure 3 diagnostics-13-02494-f003:**
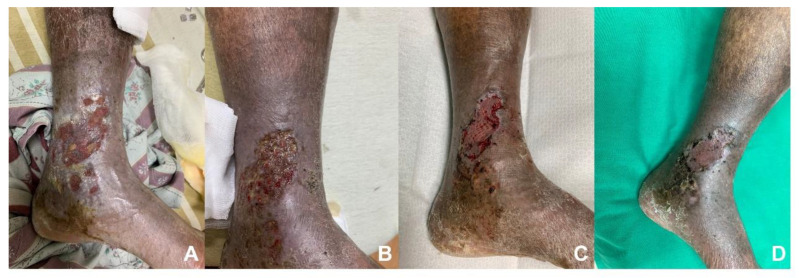
**The treatment course of venous ulcers.** (**A**–**C**) After continuous wound care with cadexomer iodine powder and intervention of varicose veins, a split-thickness skin graft was performed to improve wound healing. (**D**) Six months after the split-thickness skin graft and wound care.

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
