# Peer review of "An Overview of Clinical Examinations in the Evaluation and Assessment of Arterial and Venous Insufficiency Wounds"

_diagnostics, 2023, doi:10.3390/diagnostics13152494_

Round 1
Reviewer 1 Report
I have received for review a review article entitled “An Overview of Clinical Examinations in the Evaluation and Assessment of Arterial and Venous Insufficiency Wounds” prepared by Szu-Han Wang et al., which is being processed for publication in the journal Diagnostics (IF=3.992). The text concerns various diagnostic procedures in the area of the cardiovascular system that are important in determining the etiology of a chronic wound. The subject matter raised by the authors is extremely important because, on the one hand, cardiovascular diseases are one of the most important causes of morbidity and mortality in the world, and on the other hand, a chronic wound itself is a factor significantly limiting the quality of life and may also contribute to a life-threatening condition, e.g. in the course of an infection. The text is prepared in a generally correct, substantively correct and interesting way. It should be considered for publication in future. However, I believe that it is worth considering making a few corrections in the aspects that I present below.
1) I believe that it is worth describing the issue of the ankle-brachial index in more detail. In particular, it should be specified what limitations are associated with it, in which patient populations the result may be unreliable, and that too high an ankle-brachial index is also a significant abnormality that is associated with increased cardiovascular risk. The relationship between the ABI value and cardiovascular risk therefore takes a U-shaped form: both a lowered and too high ABI value is associated with increased cardiovascular risk.
2) I believe that in section 2.1. it is worth mentioning that reliable diagnostics in patients with lower limb ulceration in the course of lower limb arterial insufficiency is extremely important, as it opens the way to invasive procedures, primarily in the form of percutaneous angioplasty with optional stent implantation. In the same chapter, the topic of diabetes is addressed. At the same time, it is worth mentioning that a significant limitation for the effectiveness of angioplasty and stenting procedures is the risk of restenosis, and diabetes is also a risk factor for this unfavorable phenomenon. (doi.org/10.3390/ijerph182211970)
3) The presented text is a valuable summary of available diagnostic techniques. However, it returns quite a small "innovative" element in this text. Of course, the presented text is a review article, so it is not based on the results of the Authors' own research, but on the available literature. Nevertheless, the Authors could formulate what, in their opinion, is the need for further research in this field, what scientific problems in this field should be solved.
4) In my opinion, the English language is at a good level. Please review the entire text again for any minor linguistic corrections.
In my opinion, the English language is at a good level. Please review the entire text again for any minor linguistic corrections.
Author Response
Dear Reviewer,
Thanks for your valuable suggestions and comments. Our responses were attached. Please see the attachment.

Reviewer 2 Report
The authors present a well written overview over the most common diagnostic procedures in the evaluation of vascular pathology as a reason for chronic wounds.
Whereas I do have only few specific comments, I would encourage the authors to amend paragraph 2.7, respectively 2.8.
While most mentioned modalities have their fix and deserved place in the diagnostic process, there are still several shortcomings. Those are currently being addressed by newer modalities, and likely not yet broadly available technics. However, it would certainly be in interest of potential readers.
It is too challenging to depict blood supply in a dedicated anatomical region by a single phase angiography. The blood supply may be altered by collaterals, neo-angiogenesis, dilated microvessels, as well as microvascular disease. Therefore, newer Perfusion examinations are under evaluation, and might find their way to broader availability soon:
MRI Perfusion (e.g.: PMID 34441939).
CT Perfusion (e.g.: PMID 31560706, 33547479).
Review (PMID: 35522774).
-
Author Response

(The authors gave the same response as above.)

Round 2
Reviewer 1 Report
The paper has been significantly improved. I recommend it for publication in its current form.